

# A Python program to merge Sanger sequences: an update

Shiming Lin[1], Bifang Huang[2], Li-li Zhao[3], Fei Xu[4], Danni Pan[4], Xuanyang Chen[4] and Shiqiang Lin[2]

[1] School of Computing and Information Science, Fuzhou Institute of Technology, Fuzhou, Fujian, China
[2] College of Life Science, Fujian Agriculture and Forestry University, Fuzhou, Fujian, China
[3] National Key Laboratory of Intelligent Tracking and Forecasting for Infectious Diseases, National Institute for Communicable Disease Control and Prevention, Chinese Center for Disease Control and Prevention, Beijing, China
[4] College of Agronomy, Fujian Agriculture and Forestry University, Fuzhou, Fujian, China

## ABSTRACT

Gene cloning is an important step in investigating gene structure and function. To verify gene sequence, Sanger sequencing is used, which may produce several overlapping sequencing files that need to be merged before alignment to the target gene sequence is performed. Previously, we reported the Python program to Merge Sanger sequences (https://peerj.com/articles/11354/), which ran in command line and relied heavily on EMBOSS suite. In this updated version of the program, we have made several remarkable improvements. It provides a graphical user interface (GUI) written with tkinter, which is convenient and stable. It does not require users to rename the input sequences before performing merging. With regard to the implementation, the updated version utilizes Python function (Align.PairwiseAligner) to align adjacent sequences, which is more flexible (can adjust program parameter *i.e.*, the number of first-time consecutive matching bases). The new version of the program makes merging Sanger sequences much more convenient and facilitates gene study.

## INTRODUCTION

Studying gene function usually requires cloning wild-type, mutants, and insertion/deletion variants of the gene. The experimental process generally includes amplification of gene sequence from template, gel recovery, digestion with restriction enzymes, ligation to target vector, transformation of *E. coli* competent cells, growing colonies, and verification of the target gene with PCR amplification or enzymatic digestion. Plasmids with target gene need to be sent to a commercial company or core facility for sequencing verification to ensure that the gene sequence is correct, where the Sanger sequencing is used (*McGinn & Gut, 2013*; *Sanger et al., 1977*; *Sanger, Nicklen & Coulson, 1977*; *Zimmermann et al., 1988*). Each reaction of Sanger sequencing can determine more than 700 bases, thereafter the quality sequencing signal decreases. However, many genes have sequences greater than 700 in length. Therefore, it is necessary to conduct walking sequencing to determine the full-length gene sequence (*Liu et al., 2023*; *Tang et al., 2020*; *Tang et al., 2023*). In walking

Corresponding author
Shiqiang Lin, linshiqiang@fafu.edu.cn

sequencing, a sequencing primer is designed according to the previously determined sequence, and a new sequencing reaction is conducted to determine the remaining gene sequence, and so on until the full-length gene sequence is obtained.

After obtaining the result files of the sequencing reactions, the files are compared with the target gene sequence files one by one, using software such as EMBOSS needle and APE to confirm the gene sequence, segment by segment (*Davis & Jorgensen, 2022*; *Needleman & Wunsch, 1970*; *Rice, Longden & Bleasby, 2000*). However, this method is time-consuming and error-prone since multiple sequencing files are involved. Another method is to first splice all the sequences into a merged sequence according to the overlap sequence between the adjacent sequence files, then align the merged sequence with the target gene sequence (which is already known before the experiment in many cases). In this way, the researcher only has to perform alignment once, which is time-saving and not prone to errors. There is commercially expensive software for merging sequences, and academically free solutions (*Bell & Kramvis, 2013*; *Huang & Madan, 1999*). We have previously reported a free, open-source Python program, which relies on EMBOSS suite heavily (*Rice, Longden & Bleasby, 2000*). It has remarkably improved the efficiency of confirming the correct gene sequence (*Chen et al., 2021*). However, the program is not intuitive, as it runs with terminal commands and needs to rename the sequence files beforehand. In this study, we have improved the program by providing a graphical user interface (GUI) and more formats for input and output sequences (supporting fasta, seq, and txt), to make the operation more intuitive and flexible. Moreover, we use Biopython function instead of calling EMBOSS needle program, which may simplify the installation and use of our program (*Cock et al., 2009*; *Rice, Longden & Bleasby, 2000*).

Gene cloning is a basic operation in the laboratory, and it is actually the experimental starting point of many research projects. In addition, the research process often involves gene cloning, such as cloning gene mutants. This study aims to provide a convenient and efficient method for merging sequencing files in gene cloning experiments and to facilitate the study of gene function.

## METHODS

### Computer hardware and software

The software (Merge_Sanger_v3.1.py) provided in this study is based on Python and can run on Windows, macOS, and Linux computers. The hardware requirements are low and ordinary laptops are sufficient. The required Python environment is Python 3.12.1 and Biopython 1.83 (*Cock et al., 2009*). The sample sequence files for this study are available from Reference (*Chen et al., 2021*). The program, sample files, and result files of running are deposited on GitHub at https://github.com/shiqiang-lin/merge_sanger_sequences_v3.1.

### Flow of the program

The program is designed based on the characteristics of gene Sanger sequencing. During Sanger sequencing, the first sequencing reaction usually uses universal primers on the vector, and the sequencing result is obtained after the reaction. Then, in the region with a good signal, such as 200–300 bases before the end of the sequence file, the primer of

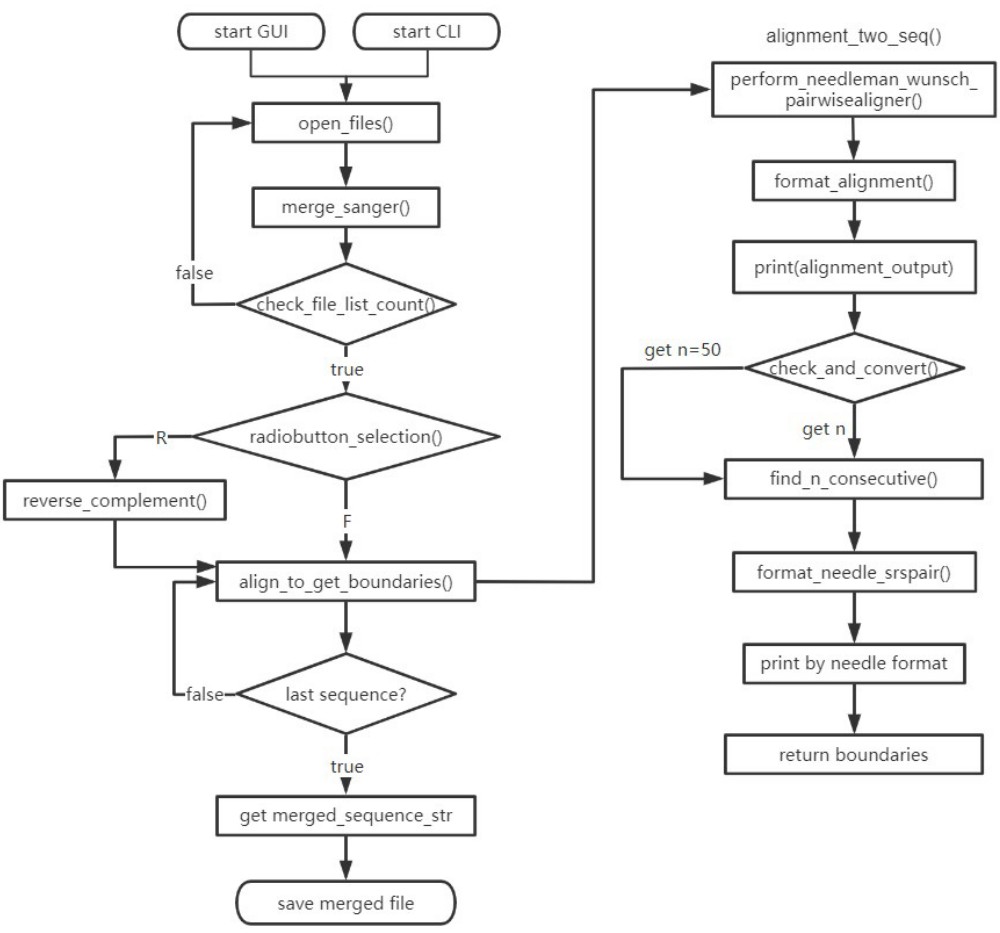

**Figure 1  Flowchart of the program.** The functions include start GUI, open main interface; open_files, add sequence files; merge_sanger, merging sequences; check_file_list_count, check the number of files added for compliance; radiobutton_selection, determine the F/R attribute of the sequence file; reverse_complement, reverse complement the sequence of R-attributes; align_to_get_boundries, call the alignment_two_seq to compare the two sequence files to obtain the alignment boundary; perform_needleman_wunsch_pairwisealigner, call Biopython's Align. PairwiseAligner to align the two sequences; format_alignment, construct and format the output string; check_and_convert, check the compliance of Consecutive Matches value; find_n_consecutive, find the positions of n matching bases in the alignment of two sequences; format_needle_srspair, save the results of sequence alignment according to the format of the needle program.

the second sequencing reaction (walking primer) is designed, so that the result of the second sequencing reaction can be merged with that of the first sequencing reaction, according to the overlapping sequences between the two results. In this way, multiple sequencing reactions can be used to obtain the full-length gene sequence. Our idea is to find the overlapping sequence between adjacent sequences, then retrieve the parts of sequences with good sequencing signal and stitch them into a full-length sequence. To better understand the business logic of the program, we have drawn a program flow, as shown in Fig. 1.

The steps in the diagram can be roughly divided into four parts, *i.e.,* input the sequences, align to get overlapping sequence, merge sequences, and output results. Before merging sequences, the program determines whether the number of input sequence files is greater than or equal to 2, then checks whether the F/R attribute is selected. After the compliance check is completed, the adjacent sequences are aligned in the order of the input, so as to obtain the boundary list of the overlapping sequence. The original input sequences are spliced according to the boundary list, and the intermediate process files are saved by the program. The resultant file recording the merged sequence will be saved by the user. With these steps, Sanger sequences can be merged quickly and smoothly.

## Run method

The program has two modes of running, GUI and CLI (see the GitHub address mentioned previously). Here, we show the GUI method with an example using macOS. The steps are described below.

(1) Create a new folder on the desktop, and you can rename the folder;
(2) Download the Merge_Sanger_v3.1.py program from the GitHub URL mentioned earlier and save it to the folder in the previous step;
(3) Open macOS Terminal, cd to the folder in the previous step, and enter the command python3.12 Merge_Sanger_v3.1.py
(4) After the program interface appears, click the "Add File" button at the top left of GUI, select the first sequence file to be merged from the pop-up window, and then select 'F' or 'R' according to the actual situation during sequencing. The 'F' indicates forward sequencing file, and 'R' indicates the reverse sequencing file. The log window on the right side of GUI prints the path and the content of input sequence. The format of the sequence file can be seq, fasta, or txt;
(5) Continue to select the second, third and other sequence files in order and select the corresponding sequencing direction (F or R) until all sequencing files have been added;
(6) Left-click the "Merge" button at the top left of GUI, and the log window on the right side displays the process of merging (the alignment results of all neighboring sequence files) and the merged sequence;
(7) Left-click the "Save Result" button at the top left of GUI to save the merged sequence, and the optional formats include seq, fasta, and txt. In the same folder as the path of the first sequence file added, a folder named 'merged_sequence plus timestamp' appears, which stores the log, the forwardized sequences (all sequence files are converted to forward sequencing result), the alignment results of adjacent sequences, and the merged sequence. These merging process files can be viewed and analyzed so that you can record and understand how the program merges the input sequences.

## Comparison with the earlier version of the program

Compared with the earlier version, the new version of the program has made large improvements. To show this, we compare the earlier and new versions of the program. The earlier version of the running process used the method in reference (*Chen et al., 2021*). The merged result file by the earlier version and the merged result file by the new version
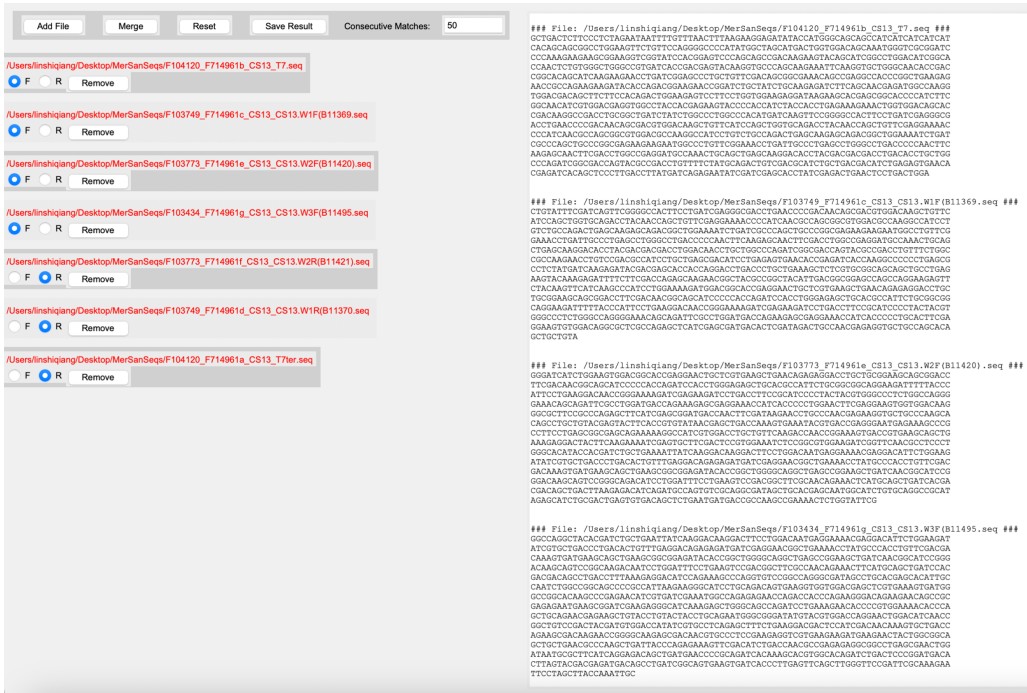

**Figure 2 Screenshot of adding sequence files.** The left side shows the added sequence files and the right log window shows the location of the added sequence files and the bases.

are aligned by EMBOSS needle (*Rice, Longden & Bleasby, 2000*), and the alignment result is viewed with TextEdit on macOS.

## RESULTS

### Program run

The program has a user-friendly GUI that is easy to use, and after starting the program from the command line, users no longer need to use the command line for the rest of the operation. The sequence files are added one by one and the forward sequencing (F) or reverse sequencing (R) are selected, as shown in Fig. 2.

In the figure, you can add a sequence file each time you click the "Add File" button, and after selecting the F or R button, you can continue to add other sequence files. During this process, the log window on the right side of the interface shows the directory and DNA sequence where each added sequencing file is located. The program requires that users only add one sequence file at a time, and select F/R for all sequences before clicking the "Merge" button, and this reduces the merging error caused by adding the wrong file.

In order to make it easier for the user to see the added sequences, the program uses an alternate highlight background when displaying the path file, so that it is not easy to make a mistake when selecting F/R, especially when the path file name is long. The order in which the files are added and the selection of F/R are very important in the process of merging sequences. The program takes these into account and is designed to help users

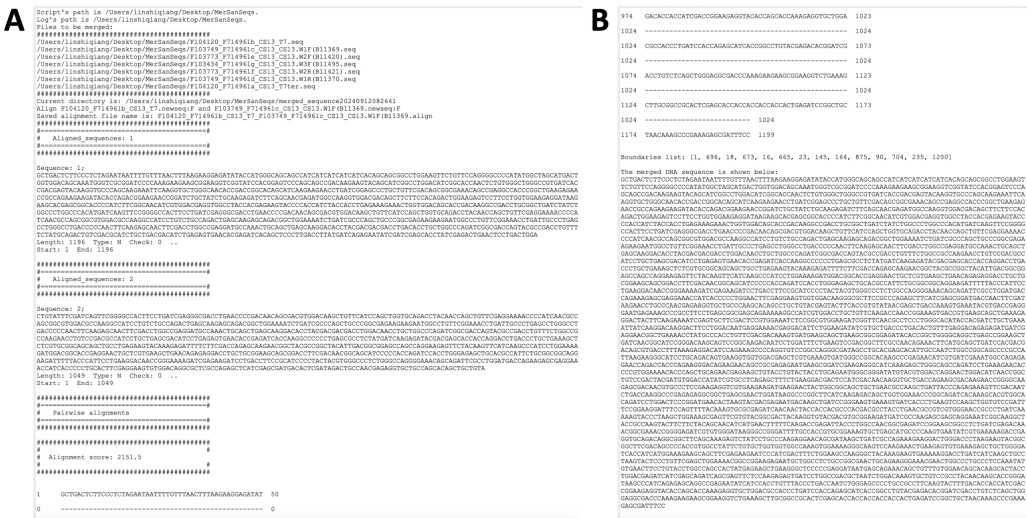

**Figure 3  Screenshots of log windows during merging sequences and saving the result.** (A) Log window after merging sequences (partial); (B) log window after saving result (partial).

add all sequence files in the correct way. Once all the sequence files have been added, users click the "Merge" button to merge all the sequencing sequences, and then click the "Save Result" button, as shown in Fig. 3.

Figure 3A shows the log window after clicking the "Merge" button, displaying all the details of the merge, including the alignment results of each of the two adjacent sequences, which is useful information for the user to get a clear picture of how the sequences are merged. Figure 3B shows the merged sequence in the log window after clicking the "Save Result" button. When saving the results, users can choose the format of the merged sequence, including seq, fasta, and txt, which may facilitate the subsequent processing of the merged sequences. After selecting the file name, path and format for the merged sequence, the result is shown in Fig. 4.

The example_input_files folder contains the original sequencing sequences used for merging. The merge_by_v3.seq is the file name (seq format) that is selected to save when the "Save Result" is executed. The merged_sequence20240912082641 folder contains the run logs, the forwardized files of all the original sequencing files, the alignment results of all adjacent sequencing files, and the merged sequences (in seq format by default).

## Comparison with earlier version of the program

Although the new version and the earlier version are based on the same principle of merging sequences, the current version has a graphical interface, and there is no need to specify the order of merging by changing the file names of the sequence files; instead, the order of added files has done the job. To compare the running results of the earlier and new versions of the program, we run the programs respectively to obtain the merged sequences,

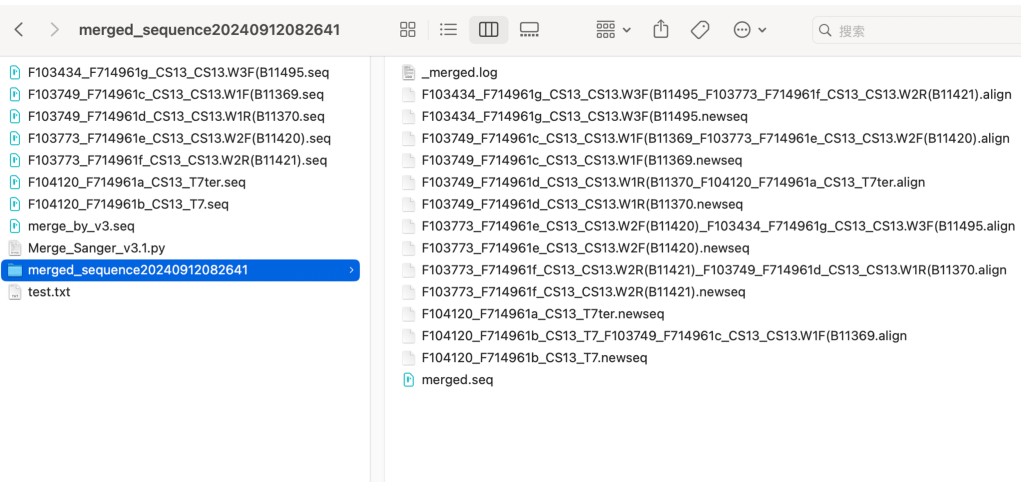

**Figure 4  The result files generated by the program.**

and then compare them with the needle program of EMBOSS suite, and the result is shown in Fig. 5.

As can be seen from the figure, the Identity is 100% and the Gaps are 0 when the sequence merge_by_v3.seq obtained by new version of the program is aligned to the sequence merged_by_v2.seq obtained by the earlier version of the program. The two sequences are the same, demonstrating that both the new version and the earlier version can get the correctly merged sequence.

## DISCUSSION

The program of this study for merging Sanger sequences has several advantages. It provides an easy-to-use GUI, which is based on tkinter, a library that comes with Python. Therefore, the GUI is stable and reliable. When operating Sanger sequence merging, except for the command that needs to be entered when opening the program, the rest can be completed with mouse clicks. There is no need to modify the file name of the original sequence files like the earlier version, simplifying the operation.

Further, the Align.PairwiseAligner method of Biopython is used to conduct the alignment of adjacent sequences, which is simpler than the earlier method of calling the EMBOSS needle program. The needle program reads the input sequence files, saves the alignment sequence file, and then parses the alignment sequence file in the earlier version of the program. The Align.PairwiseAligner method makes it possible to find matching bases for adjacent sequences in memory.

Moreover, the new version of the program can adjust the number of consecutive matching bases that appear for the first time between adjacent sequences, improving the flexibility of the program. The principle is that the signal in the middle of the sequencing reaction is good, while that in the two ends is poor. For two adjacent sequences, when the bases of the previous sequence and the latter sequence are continuously paired more than 50, the previous sequence is approaching to the end of the sequencing reaction, where

```
####################################
# Program: needle
# Rundate: Thu 12 Sep 2024 08:52:55
# Commandline: needle
#    [-asequence] merge_by_v2.seq
#    [-bsequence] merge_by_v3.seq
#    -outfile compare.needle
# Align_format: srspair
# Report_file: compare.needle
####################################

#=======================================
#
# Aligned_sequences: 2
# 1:
# 2:
# Matrix: EDNAFULL
# Gap_penalty: 10.0
# Extend_penalty: 0.5
#
# Length: 4411
# Identity:     4411/4411 (100.0%)
# Similarity:   4411/4411 (100.0%)
# Gaps:            0/4411 ( 0.0%)
# Score: 22055.0
#
#
#=======================================

    1 GCTGACTCTTCCCTCTAGAATAATTTTGTTTAACTTTAAGAAGGAGATAT     50
      ||||||||||||||||||||||||||||||||||||||||||||||||||
    1 GCTGACTCTTCCCTCTAGAATAATTTTGTTTAACTTTAAGAAGGAGATAT     50

   51 ACCATGGGCAGCAGCCATCATCATCATCATCACAGCAGCGGCCTGGAAGT    100
      ||||||||||||||||||||||||||||||||||||||||||||||||||
   51 ACCATGGGCAGCAGCCATCATCATCATCATCACAGCAGCGGCCTGGAAGT    100

  101 TCTGTTCCAGGGGCCCCATATGGCTAGCATGACTGGTGGACAGCAAATGG    150
      ||||||||||||||||||||||||||||||||||||||||||||||||||
  101 TCTGTTCCAGGGGCCCCATATGGCTAGCATGACTGGTGGACAGCAAATGG    150
```

**Figure 5** Screenshot of comparing the results of earlier and new versions of the program (partial).

the sequencing signal gradually deteriorates. However, the signal of the latter sequence is better at this time, since the signal is better in the middle of reaction. The program gets the sequence before the continuous matching region of the previous sequence, the sequence of the continuous matching region, and the sequence after the continuous pairing region of the next sequence, which is the biochemical basis and the idea of the sequence merging procedure.

Then consider this example. The forward sequencing reactions have determined part of the sequence *via* walking, and reverse sequencing also have determined part of the sequence *via* walking the reverse direction. The forward and reverse sequencing overlaps somewhere in the middle of the gene sequence. If the overlapping area is less than 50 bases, then the earlier version of the program will not work. In the new version, users can merge the two adjacent sequence files first by modifying the "Consecutive Matches" parameter (*e.g.*, down to 35 or 40). Then users can use the resulting sequence file to merge with the rest of the sequence files.

## CONCLUSIONS

In this study, we provide a free, open-source method for merging Sanger sequences. The GUI program can help researchers merge multiple sequence files to confirm the correct gene sequence cloned, and facilitate the study of gene function.

### Funding

This study is funded by the Spark Project of Fujian Provincial Department of Science and Technology (No. 2023S0012). The funders had no role in study design, data collection and analysis, decision to publish, or preparation of the manuscript.

### Grant Disclosures

The following grant information was disclosed by the authors:
The Spark Project of Fujian Provincial Department of Science and Technology: 2023S0012.

### Competing Interests

The authors declare there are no competing interests.

### Author Contributions

- Shiming Lin conceived and designed the experiments, performed the experiments, analyzed the data, prepared figures and/or tables, authored or reviewed drafts of the article, and approved the final draft.
- Bifang Huang performed the experiments, analyzed the data, prepared figures and/or tables, and approved the final draft.
- Li-li Zhao performed the experiments, analyzed the data, prepared figures and/or tables, and approved the final draft.
- Fei Xu performed the experiments, prepared figures and/or tables, and approved the final draft.
- Danni Pan performed the experiments, prepared figures and/or tables, and approved the final draft.
- Xuanyang Chen performed the experiments, prepared figures and/or tables, and approved the final draft.
- Shiqiang Lin conceived and designed the experiments, performed the experiments, analyzed the data, prepared figures and/or tables, authored or reviewed drafts of the article, and approved the final draft.

### Data Availability

  The program source code, test gene sequence files, and test report are available at GitHub and Zenodo:
    - https://github.com/shiqiang-lin/merge_sanger_sequences_v3.1
    - shiqiang-lin. (2024). shiqiang-lin/merge_sanger_sequences_v3.1: v3.1 (v3.1). Zenodo. https://doi.org/10.5281/zenodo.13766670.

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
