# Peer review of "A Python program to merge Sanger sequences: an update"

_PeerJ, doi:10.7717/peerj.18363_

## Round 0.1 · original submission · Major Revisions

Dear Dr. Lin and colleagues:

Thanks for submitting your manuscript to PeerJ. I have now received two independent reviews of your work, and as you will see, the reviewers raised some concerns about the research. Despite this, these reviewers are not recommending rejection, although there is a lack of enthusiasm for your work. In particular, the update to your program does not seem to be robustly tested, and there appear to be multiple problems with platform-wide installation and execution. Thus, I encourage you to revise your manuscript and methodology, accordingly, considering all concerns raised by the reviewers.

I look forward to seeing your revision, and thanks again for submitting your work to PeerJ.

Good luck with your revision,

-joe

Reviewer 1 ·

Basic reporting

OK

Experimental design

The experimental design is far too simple. The authors do 1 test. What happens when there are sequencing errors? How about artificially mutating the sequences and observing when merges fail? This would help tune the match/mismatch parameters, for example. Of course, if you were doing lots of runs you would rather have the CLI than the GUI. So maybe have a CLI interface also.

Validity of the findings

The idea that the new program is faster than the old one because function calls are faster than system calls is not well-reasoned. EMBOSS programs are written in C. As a general observation, C programs run about 100x faster than Python. The old version should be faster than the new one. That might not be true in practice because EMBOSS needle is the classic Needleman-Wunsch algorithm and BioPython might be doing something else (at which point the authors might mention that the algorithms are not the same).

Having to install EMBOSS is not a big deal. It takes just as much effort to install EMBOSS as it does BioPython.

Cite this review as

Reviewer 2 ·

Basic reporting

The manuscript was well written in English and well structured with relevant results.

Experimental design

no comment

Validity of the findings

no comment

Additional comments

This paper demonstrated a major improvement of Sanger sequence Merger application that may have general interest to gene research. Recommend to accept with following aspects that could be improved:

1. "METHODS--Computer hardware and software" part:
The application depends not only python3.12 and biopython-1.83, but also numpy which not always gets automatically installed together with biopython. For example when testing on Window10, numpy-2.0.1 was installed together with biopython-1.83 with pip install, but not on our HPC system with Rocky8 Linux, and I have to install numpy with conda separately, it would be more accurate if the author could list all the dependencies of the application for installation.

2. "Run method" part: "python3.12 Merge_Sanger_v3.py" Not always works, but "python Merge_sanger_v3.py" in general works when given the require environment.

3. The current script for tkinter GUI works on local desktop like Windows10, but not with X11 forwarding when opening GUI from a remote Linux server. In order to enable that, line 24 needs to be changed to: master.attributes('-zoomed', True)
But then, local desktop GUI will be disabled with the change. It would be ideal if the script could sense the need of X11 forwarding or specify the limit of current version that only recommended to run on local desktop/laptop.

4. Although this is an updated version of Merge_Sanger_v2 and the author compared the v3 application with the older version, it would further justify the significance of the paper if the comparison could include recently published applications like sangeranalyseR (https://academic.oup.com/gbe/article/13/3/evab028/6137837)

Cite this review as

---

## Round 0.2 · accepted · Accept

Dear Dr. Lin and colleagues:

Thanks for again revising your manuscript. I now believe that your manuscript is suitable for publication. Congratulations! I look forward to seeing this work in print, and I anticipate it being an important resource for the bioinformatics field. Thanks again for choosing PeerJ to publish such important work.

Best,

-joe

Reviewer 2 ·

Basic reporting

No comment

Experimental design

No comment

Validity of the findings

No comment

Additional comments

All reviewers’ questions have been thoroughly answered and concerns has been well addressed. The manuscript and Merge_sanger_v3.1 have been further improved so that reaching PeerJ’s standard of publication. Recommend accepting as is.

Cite this review as